# Beneficial Metabolic Effects of Chitosan and Chitosan Oligosaccharide on Epididymal WAT Browning and Thermogenesis in Obese Rats

**DOI:** 10.3390/molecules24244455

**Published:** 2019-12-05

**Authors:** Jin Wang, Wanping He, Di Yang, Hua Cao, Yan Bai, Jiao Guo, Zhengquan Su

**Affiliations:** 1Guangdong Engineering Research Center of Natural Products and New Drugs, Guangdong Provincial University Engineering Technology Research Center of Natural Products and Drugs, Guangdong Pharmaceutical University, Guangzhou 510006, China; 15671620448@163.com (J.W.); wj15013060744@163.com (W.H.); 18687566544@163.com (D.Y.); 2Guangdong Metabolic Diseases Research Centre of Integrated Chinese and Western Medicine, Guangdong TCM Key Laboratory for Metabolic Diseases, Key Laboratory of Modulating Liver to Treat Hyperlipemia SATCM, Level 3 Laboratory of Lipid Metabolism SATCM, Institute of Chinese Medicinal Sciences, Guangdong Pharmaceutical University, Guangzhou 510006, China; 3School of Chemistry and Chemical Engineering, Guangdong Pharmaceutical University, Zhongshan 528458, China; caohua@gdpu.edu.cn; 4School of Public Health, Guangdong Pharmaceutical University, Guangzhou 510310, China; angell_bai@163.com

**Keywords:** chitosan, chitosan oligosaccharide, obesity, thermogenesis, white adipose tissue, brown adipose tissue

## Abstract

Many anti-obesity chemicals have been withdrawn from the market due to serious adverse reactions, and the researchers have turned their attention to low-toxic natural products. Previous studies have demonstrated that chitosan (CTS) and chitosan oligosaccharide (COS) were low-toxic natural products for the use of weight loss. However, it is still unclear whether CTS and COS have positive effects on the thermogenesis. In this study, CTS and COS significantly reduced the weight gain of rats without affecting food intake and effectively inhibited adipose tissue hypertrophy and hyperplasia. Consistently, CTS and COS significantly increased the thermogenic capacity of obese rats induced by high-fat diet (HFD) and increased the expression of browning genes and proteins (UCP1, PGC1α, PRMD16, and ATF2) in white adipose tissue (WAT) and brown adipose tissue (BAT). In vitro, COS inhibited the formation of mature adipocytes and increased the expression of browning genes. In conclusion, COS and CTS was used to explore the function and mechanism on thermogenesis, and CTS and COS can increase the browning of WAT and the thermogenesis of BAT to inhibit obesity. This effect may be achieved by promoting the expression of browning and thermogenic genes, providing new ideas for the utilization of COS and CTS.

## 1. Introduction

Obesity is an increasing burden on the human health care system, and the effective control and treatment of obesity can significantly decrease the incidence of other related metabolic syndromes, such as type 2 diabetes, cardiovascular diseases, and other serious disorders [1]. Currently, US Food and Drug Administration (FDA)-approved anti-obesity drugs mainly include: Orlistat, lorcaserin, phentermine, phentermine/topiramate extended-release capsules (Qsymia), naltrexone/bupropion sustained-release tablets (Contrive), and liraglutide injection [2,3]. Although these drugs have significant effects on the treatment of obesity, the FDA has warned about these drugs because of their serious adverse reactions and side effects in the cardiovascular, central nervous system, and gastrointestinal tract. These side effects and adverse reactions also limit their sales and use in the market, and bring greater medication pressure to patients [3]. Orlistat is an inhibitor of gastrointestinal lipase. Compared with the other five anti-obesity drugs, orlistat has fewer side effects, and it is currently the only authorized supplemental anti-obesity drug in many countries and regions [4].

Energy homeostasis, including glucose homeostasis, is largely dependent on the activity and function of adipose tissue. There are two major types of adipose tissue in mammals: Brown adipose tissue (BAT) and white adipose tissue (WAT). WAT primarily serves as energy storage and secretes hormones in response to nutritional signals. BAT, including classical BAT and the newly identified beige adipose tissue, functions to resist cold and obesity through adaptive heat generation, namely, thermogenesis, in processes mainly mediated by the expression and activation of mitochondrial uncoupling protein 1 (UCP1) [5]. Thermogenic adipocytes, including brown adipocytes and beige fat cells, are prolific in human infants and adults, attracting significant interest as potential targets for increasing energy consumption and combating obesity [6].

Brown adipocytes primarily rely on UCP1, which reduces the proton gradient by uncoupling the respiratory chain and catalyzes the synthesis of cyclic adenosine monophosphate (AMP) (cAMP) in mitochondria to stimulate heat-generating capacity [7]. However, brown adipocytes do not have the ability to mediate adaptive thermogenesis except through UCP1 [8]. UCP1 activation, therefore, represents a potential protective mechanism against obesity, diabetes, and dyslipidemia. The first step of the activation of brown adipocytes is the release of norepinephrine by the sympathetic nervous system at the surface of brown adipocytes. Subsequently, adrenergic receptors are recruited, and then cAMP sharply increases and activates protein kinase A (PKA), which induces lipolysis. Released free fatty acids (FFAs) reverse the inhibitory effects of purine nucleotides on UCP1 and activate UCP1 in the inner mitochondrial membrane. Activated UCP1 rapidly induces the return of protons to the mitochondrial matrix and consequently decreases hydrogen levels, leading to the activation of respiratory pathways and fatty acid oxidation [9,10]. Under such conditions, the proton pathway is not used to phosphorylate ADP to produce heat [9]. Studies have shown that adult human BAT contains beige fat cells, and the elucidation of the signaling pathways that improve thermogenesis in BAT may provide new potential therapeutic opportunities for obesity or for people with low levels of active BAT [10].

Peroxisome proliferator-activated receptor γ coactivator-1α (PGC-1α) is the critical moderator of mitochondrial biogenesis in brown and beige adipocytes. Interestingly, the ectopic expression of PGC-1α induces the browning of white adipocytes via the increased expression of UCP1 and peroxisome proliferator-activated receptor-gamma (PPARγ)/RXR [11]. Additionally, PRD1-BF1-RIZ1 homologous domain-containing 16 (PRDM16) regulates the activity of PGC-1α, PPAR, and CCAAT/enhancer binding proteins (CEBPs), and controls the differentiation of both brown and beige fat [12,13,14]. In addition, activating transcription factor 2 (ATF2) is a conserved DNA response element in the PGC-1α promoter [15]. Taken together, all of these transcriptional factors, including PGC-1α, PRDM16, and ATF2, lead to thermogenesis in brown and beige fat through UCP1-mediated mechanisms. The synergic actions and negative feedback among these factors and presence of simultaneous thermogenic signaling pathways in the body has a causal relationship remain unknown; we hypothesize that these pathways have common connections.

Chitosan (CTS) and its derivatives are products derived from the reprocessing of aquatic products such as shrimp shells and crab shells, and CTS is stored in large quantities in nature [16]. Chitosan oligosaccharide (COS) is an oligosaccharide obtained by the hydrolysis of CTS [17]. COS has attracted much attention due to its small molecular weight, good water solubility, and diverse biological activities, such as antibacterial, anti-inflammatory, antioxidant, antiobesity, and antidiabetic effects [17,18].

Recently, the development of COS supplements as antiobesity and antidyslipidemia therapeutics has attracted the interest of many researchers. In fact, researchers have proposed a variety of mechanisms to explain the antiobesity and lipid-lowering activities of COS. For example, COS has been found to inhibit lipase and bile acids activity, reduce fat absorption, and increase fat excretion [19,20]. Notably, COS improved obesity and blood lipid levels by mediating the PPAR oxidative phosphorylation pathway in the mouse brain and stomach [18,21].

Based on previous laboratory research, COS with an average molecular weight not exceeding 1 KDa (COST), COS with an average molecular weight not exceeding 3 KDa (COSM), and CTS were used to treat high-fat diet (HFD)-induced obese rats to explore their antiobesity activities and the mechanisms of the browning of epididymal WAT and of thermogenesis.

## 2. Results

### 2.1. COST, COSM, and CTS Reduce the Weight Gain and Increase Energy Expenditure of Obese Rats

After eight weeks of drug treatment, the weight gain trends of the COST-, COSM-, CTS-, and orlistat-treated groups were slower than that of the HF group (body weight: HF, 712.31 ± 17.00 g; orlistat, 618.98 ± 11.38 g; COST, 620.87 ± 19.34 g; COSM, 630.29 ± 20.51 g; CTS, 616.98 ± 21.99 g) (weight gain: HF, 102.03 ± 26.45; orlistat, 69.07 ± 20.88; COST, 52.78 ± 20.21; COSM, 55.78 ± 22.19; CTS, 60.04 ± 24.75) (Figure 1B,C and Appendix A) and the food intake of the rats was not affected (Figure 1A and Appendix A). These results indicate that the suppressed weight gain was not due to a decrease in food intake. To determine whether COST, COSM, and CTS inhibited weight gain due to a decrease in adipose tissue, anatomically obtained fat was weighed, and the fat content of the rat was calculated. Consistent with the decrease in body weight gain, the fat content results showed a significant reduction in the fat content of the COST, COSM, and CTS groups and the orlistat group compared to that of the HF group (Figure 1D). By weighing different types of fat, we also found that the brown fat content of the COST, COSM, and CTS groups was significantly increased (Appendix A). Orlistat inhibits fat absorption by inhibiting lipase activity, which resulted in a decrease in body weight and WAT in rats. This study found that COST, COSM, and CTS are equivalent to orlistat in reducing body weight and WAT. Compared with the HF group, the orlistat and COST, COSM, and CTS groups had significantly reduced fat cell content, and the fat cells were uniform and small in size (Figure 1E). Oral glucose tolerance tests (OGTTs) showed that the blood glucose levels in the COST, COSM, and CTS treatment groups were lower than those in the HF group, showing a significantly smaller area under the curve (AUC), suggesting that COST, COSM, and CTS may improve glucose metabolism in rats (Figure 1F).

To determine whether COST, COSM, and CTS reduce body weight gain by regulating energy expenditure and to explore how these treatments induce heat production, we measured the rectal temperature of the rats at room temperature (25 °C) weekly and the temperature compensation at short-term cold stimulation (4 °C) before sacrifice. At room temperature (25 °C), the rectal temperatures of COST, COSM, CTS, and orlistat-treated Sprague Dawley (SD) rats were significantly increased compared with those of HFD rats (Figure 1G), suggesting an increase in thermogenesis in obese SD rats. After 6 h of cold exposure, the rate of decrease in rectal temperature was slower in SD rats treated with COST, COSM, and CTS than in that of HFD rats, and the rate of decrease was significantly lower in the COST group than in the HF group, further suggesting that COST, COSM, and CTS increased the heat production of obese rats (Figure 1H). To further verify whether COST, COSM, and CTS affect the thermogenesis of obese rats, the O2 consumption and CO2 release of obese rats were determined. The results showed that COST, COSM, and CTS increased oxygen consumption and carbon dioxide release in obese rats (Figure 2A,B) and increased respiratory entropy (RER) and heat production in obese rats (Figure 2C,D). These results indicate that COST, COSM, and CTS promote energy release in obese rats.

### 2.2. COST, COSM, and CTS Improve Serum Levels and HFD-Induced Fatty Liver in Obese Rats

To investigate the effects of COST, COSM, CTS, and orlistat on blood lipid levels in rats, we measured serum total cholesterol (TC), triglyceride (TG), high-density lipoprotein-cholesterol (HDL-C), low density lipoprotein-cholesterol (LDL-C), free fatty acid (FFA), and glucose levels in rats (Appendix A). TG, TC, and LDL-C levels were significantly decreased (Figure 3A–D), HDL-C levels were significantly increased (Figure 3C), and glucose and FFA levels were significantly decreased (Figure 3E,F) in the orlistat-, COST-, COSM-, and CTS-treated groups compared with the HF group, suggesting that COST, COSM, CTS, and orlistat can exert certain hypoglycemic and hypolipidemic effects. We also found that the insulin in the serum of the treatment group was significantly higher than that of the HF group (Figure 3G).

The anatomical analysis of the liver of SD rats showed that the livers of the drug-administered group and the NF group were normal and had a vermilion color without fatty liver development. The HF group showed larger fat granules, the texture was rough, and the color was obviously yellowish (Figure 3H). These results indicate that COST, COSM, CTS, and orlistat may protect the liver from the steatosis induced by a HFD. Interestingly, the effects of COST, COSM, and CTS are comparable to orlistat. In addition, hematoxylin and eosin (HE) staining of liver pathological sections also showed the same results. Compared with the NF group, the liver cells of the HF group were deformed, the size was not uniform, and there were more fat vacuoles; the fat vacuoles in the liver of the orlistat, COST, COSM, and CTS treatment groups were significantly reduced compared to the HF group (Figure 3I).

### 2.3. Differential Gene Expression (DGE)

To ensure the reliability of the data analysis results, low quality, joint contamination reads and reads with unknown base N content were excluded. The clean reads obtained after filtering the data were aligned to the reference gene sequence to obtain the alignment ratio. The gene sequence results showed that the ratio of filtered genomes was greater than 70% (Appendix A), and it is generally believed that a genomic alignment rate of 60% indicates that the sample quality is reliable. To reflect the correlation between the gene expression of samples, the Pearson correlation coefficient for all gene expression levels between each sample was computed. The more positive the correlation coefficient was, the more similar the gene expression levels were. The sample correlation heat map results (Figure 4A) showed that the correlation coefficient of each group was close to 1, indicating that the biological repeatability was good.

Principal component analysis (PCA) and cluster analysis were utilized to compare differences between samples to find outlier samples and to identify sample clusters with high similarity. The results showed that the samples of the HF group showed higher outliers compared to the other groups (Figure 4B). The tree diagram above the heat map (cluster analysis of the columns) shows that the differential genes of COST vs. model, COSM vs. model, and CTS vs. model were the same as those of control vs. model (up-regulated and down-regulated genes were consistent, red means up-regulation and blue means down-regulation) (Figure 4E). It can be observed that these differences are most evident in the control vs. model, indicating that COST, COSM, and CTS can inhibit these genetic differences (Figure 4E). PCA and the clustered heat map of DEGs showed that there were differences in the DEGs among the COST, COSM, CTS, and HF groups. Compared with the HF group, the number of upregulated genes was 167, 237, and 294, and the number of downregulated genes was 283, 347, and 322 in the COST, COSM, and CTS groups, respectively (Figure 4C). The DEGs in each experimental group were analyzed with a Venn diagram (Figure 4D): 64, 154, and 170 differential genes were found in the COST, COSM, and CTS groups, respectively. These results indicated that COST, COSM, and CTS can regulate gene transcription levels in obese rats (Figure 4D).

The results of the Gene Ontology (GO) annotation (Figure 5A) and enriched bubble map analysis (Figure 5B) of the differential genes indicated that the differential genes of obese rats mainly involved biological processes (blue) and cellular components (red). The impacted cellular components were cell membrane components (such as several plasma members), and the biological processes were mainly associated with the development of multicellular organisms (such as system development, multicellular organism development, anatomical structure development, developmental process, animal organ development, and regulation multicellular organismal development). The Kyoto Encyclopedia of Genes and Genomes (KEGG) metabolic pathway is divided into seven branches: Cellular processes, environmental information processing, genetic information processing, human disease, metabolism, organismal systems, and drug development. The results of the KEGG pathway annotation and enriched bubble map analysis indicated that the differential genes in obese rats were concentrated in environmental information, disease, and metabolism (Figure 5C), including cellular responses and signaling in metabolic diseases, lipid metabolism, and energy metabolism (Figure 5D), many of these pathways are associated with lipid metabolism and thermogenesis.

DGE analysis showed that genes involved in fat synthesis, metabolism, and energy metabolism were differentially expressed between rats fed a HFD and NF diet. Therefore, we speculate that CTS, COST, and COSM may reduce the body weight of SD rats by affecting metabolic genes.

### 2.4. COST, COSM, and CTS Induce the Browning of Epididymal WAT

Based on the gene sequencing results, we selected genes related to lipid metabolism and energy metabolism for verification, including a browning marker gene and thermogenic genes, namely, PRDM16, PGC1α, UCP1, and P38-MAPK. The results showed that the mRNA and protein levels of UCP1, PRDM16, and PGC-1α in the epididymal fat of SD rats in the COSM, COST, and CTS treatment groups were significantly higher than those in the HF group (Figure 6A–C). Based on these results, we can preliminarily conclude that COSM, COST, and CTS may promote WAT browning in the epididymis of obese SD rats.

To investigate whether the browning of white fat is promoted by the activation of UCP1 via the PKA/P38-MAPK pathway, we examined the mRNA and protein expression of P38-MAPK. We found that compared with the HF group, the COST, COSM, and CTS treatment groups showed significantly increased mRNA and protein expression levels of P38-MAPK (Figure 6A–C). To verify beige fat production, the transcription of the Slc27a1 and TMEM26 genes was quantified. The results showed that the expression of the Slc27a1 and TMEM26 genes was significantly increased in the COST, COSM, and CTS treatment groups compared with the HF group (Figure 6D).

### 2.5. COST, COSM, and CTS Promote BAT Thermogenesis

To explore changes in BAT in obese SD rats after treatment administration, we examined the expression of thermogenic genes and their translated proteins in BAT. These results were identical to the expression of these genes in epididymal WAT; the mRNA and protein expression levels of UCP1, PRDM16, and PGC-1α in brown fat were significantly higher in obese SD rats treated with COST, COSM, and CTS than in those only administered the HFD (Figure 7). In addition, the gene and protein expression levels of P38-MAPK and downstream ATF2 were significantly increased in obese SD rats in the COST, COSM, and CTS groups (Figure 7). Under the premise of inhibiting weight gain and promoting an increase in BAT, COST, COSM, and CTS may promote BAT formation and thermogenesis.

### 2.6. COST and COSM Affect 3T3L1 Preadipocyte Differentiation

The appropriate concentrations of COST and COSM treatment in the preliminary screen were 0.5 mg/mL, 2.5 mg/mL, and 5 mg/mL, and the time to induce differentiation in 3T3-L1 cells was 15 days (Appendix A). The treatment group was treated with COST and COSM while inducing cell differentiation. The cells induced by the medium were stained with oil red O. The staining results showed that the COSM and COST treatments effectively inhibited the formation of lipid droplets in a concentration-dependent manner (Figure 8A,B). To explore the effects of COST and COSM on the lipolysis of 3T3-L1 adipocytes, the TG content of mature adipocytes was determined. The experimental results showed that compared with the model condition, COST and COSM effectively promoted the lipolysis of TG in 3T3L1 cells (Figure 8C). Furthermore, the glucose consumption capacity of adipocytes was calculated by measuring the glucose content in the medium, thereby evaluating the cell glucose consumption capacity. The results showed that both the COST and COSM treatment could promote the glucose consumption of 3T3L1 preadipocytes (*p* < 0.05) in a concentration-dependent manner (Figure 8D).

The mRNA and protein expression levels of UCP1, PRDM16, and PGC-1α in the high-dose COST and COSM treatment groups were significantly higher than those in the model group (Figure 8E–G). The median and low-dose COST and COSM treatment groups also had a certain increase in the mRNA and protein expression of UCP1, PRDM16, and PGC-1α (Figure 8E–G), but some results were not statistically significant. These results further validate the results of the in vivo experiments. COST and COSM may promote the expression of thermogenic genes and browning genes such as UCP1, PGC1α, and PRAM16, thereby promoting WAT browning and thermogenesis.

## 3. Discussion

The potential application value of BAT in human metabolism is very large. Recent reports have demonstrated that the activation of BAT can reduce low serum TG and cholesterol levels in obese patients and improve diet-induced metabolic diseases such as atherosclerosis, fatty liver, diabetes, and obesity [22,23]. The regulation of WAT browning and BAT-activated heat production are key targets for anti-obesity treatment. Increasing numbers of researchers are working to identify natural compounds that can stimulate browning. We previously confirmed the efficacy of CTS and its degradation product COS in the treatment of obesity and hyperlipidemia [24,25,26]. Consistent with the body weight results, we observed that the expression of the obesity-associated gene FTO was reduced in CTS-, COST-, and COSM-treated rats compared with HF rats (Appendix A).

Here, we provide sufficient evidence to demonstrate the anti-obesity effects of CTS, COST, and COSM and their ability to induce WAT browning. Many anti-obesity drugs have a certain negative impact on appetite, defecation, and the central nervous system [27]. However, CTS, COST, and COSM can decrease weight gain without suppressing appetite. This anti-obesity effect is achieved by reducing the fat content. The thermogenic capacity of the obese group was lower than that in the normal weight group, mainly due to the reduction in BAT content in obesity [28]. Our results are consistent with this finding; CTS, COST, and COSM increased the BAT content. Under normal temperature and cold stimulation conditions, rats in the CTS, COST, and COSM treatment groups had higher anal temperatures than rats in the HF group, indicating stronger thermogenic capacity. The evidence for the changes in the metabolic indicators volume of oxygen (VO_2_), volume of carbon dioxide (VCO_2_), respiratory exchange ratio (RER), and heat production further confirmed that CTS, COST, and COSM increased energy expenditure. Most people with metabolic diseases, especially obesity, are usually accompanied by high levels of serum lipids. More than 90% of people with severe obesity have nonalcoholic fatty liver disease (NAFLD) [29]. Similarly, our results indicated an improvement in serum lipid levels by CTS, COST, and COSM, which was consistent with our previous studies [20,24].

Previous studies have shown that CTS mainly promotes excretion by physically entangling fat in the intestines to achieve anti-obesity effects [30]. To investigate whether CTS, COST, and COSM can exert anti-obesity effects by affecting transcription levels, we conducted gene sequencing on fat samples. The results of DGEs analysis showed that COST, COSM, and CTS improve obesity by regulating gene transcription levels. Based on the DEGs involved in energy and lipid metabolism, we found some interesting genes or pathways, including the MAPK pathway, the PPAR signaling pathway, and other thermogenic and browning pathways.

CTS, COST, and COSM promote liver PPARα gene expression in the liver (Appendix A), which may increase liver fatty acid oxidation and reduce TG levels to inhibit fat hyperplasia and hypertrophy [31]. UCP1 is considered a key gene for nonshivering heat production and is also considered a hallmark gene for WAT brown fat. Studies have shown that the secretion of a sufficient amount of PRDM16 in WAT to bind to PGC1α promotes PGC1α transcription, stimulates WAT browning, and promotes BAT production [32]. Our results suggest that CTS, COST, and COSM may activate UCP1 in a PKA-MAPK-dependent manner and in a PKA-MAPK-independent manner (Appendix A) [33]. In addition, the lipid metabolism genes, Slc27a1 and TMEM26, which are specifically expressed in beige fat [34], are also increased with CTS, COST, and COSM treatment. The increased expression of these genes indicates an increase in WAT browning and thermogenesis, which become key strategies for addressing obesity.

However, as a gene that regulates fat synthesis, researchers usually achieve anti-obesity effects by inhibiting Dio2 and PPARγ [35]. We found that the expression of Dio2 and PPARγ was increased in the epididymal fat of CTS-, COST-, and COSM-treated rats compared with the rats in the HF group (Appendix A). Another argument is that Dio2 and PPARγ also play an important role in promoting browning [36], as PPARγ is essential for maintaining WAT and BAT differentiation [37] and Dio2 activation promotes adrenergic response and energy expenditure [38].

In the process of inducing the differentiation of 3T3-L1 preadipocytes into mature adipocytes in vitro, glucose in the medium was finally synthesized into TG through the de novo lipogenesis pathway [39]. This process also occurs in body fat accumulation in many cases. COST and COSM reduce intracellular TG and extracellular glucose levels, thereby reducing the formation of intracellular lipid droplets.

It is generally believed that the degree of polymerization (DP), the degree of deacetylation (DD), and the molecular weight (MW) are the principal characteristics and key factors of COS [17,40,41]. COS with a higher DP (≥6), a lower DD, and a lower MW has a stronger biological activity than COS with a lower DP, a higher DD, and a higher MW [17,41]. The DP of COS with glucosamine as the monomer is 2–20, and the maximum MW of COS is less than 4000 [17]. The MW of CTS is generally higher than 4000. Theoretically, the MW of COS with the strongest biological activity is 1000–3000. Therefore, this study selected COST (MW ≤ 1000), COSM (MW ≤ 3000), and CTS (MW > 4000) as the research objects and their DDs are higher than 90%. COST, COSM, and CTS represent three molecular weight COS derivatives, which researchers focus more on. However, the anti-obesity effects of COST, COSM, and CTS in this study did not show significant differences. We conclude that the reasons for this result are mainly: (1) The MW ranges of the two COSs and CTS in this experiment had a region of common overlap; (2) the DP of COSs and CTS used in this study was not and cannot be controlled; (3) the active substances or groups that are effective in COST, COSM, and CTS may be the same or similar. The complex structure of COS determines the difficulty of its biological evaluation, and it is one-sided and difficult to evaluate its anti-obesity activity based only on the molecular weight of COS. Therefore, the next step may be to study the specific anti-obesity effect of COSs with specific DP.

Most importantly, we demonstrated that CTS, COST, and COSM accelerate the browning of WAT and the thermogenesis of BAT by upregulating the gene expression of UCP1, PRDM16, and PGC-1α, which accelerates energy metabolism in obese rats. In addition, CTS, COST, and COSM also upregulate p38-MAPK gene expression, which is an important target for controlling energy metabolism. In summary, the results of the current work indicate that CTS, COST, and COSM may be promising natural products for the prevention and treatment of obesity.

Previous anti-obesity effects on COS and CTS were primarily focused on their effects on leptin resistance and fat formation [21,25,42]. Recent research reports show that browning of WAT and increased thermogenesis are effective against obesity [43,44], and there are few studies on COS and CTS in promoting WAT browning and increasing thermogenesis. This study mainly explored the role and specific mechanism of COS and CTS to promote WAT browning and increase thermogenesis. Nowadays, the potential side effects or adverse drug reactions of anti-obesity chemicals on the market have grown to be a major public health problem and a major obstacle to the development of new drugs [45]. Therefore, natural products with low toxicity and anti-obesity effects have become popular trends in anti-obesity drugs or supplements. As a safe and non-toxic anti-obesity natural product, CTS and COS have many advantages, such as wide source, high yield, good biocompatibility, and are economical and practical [17,46,47]. This study further clarified the role and mechanism of COS and CTS against obesity. By comparing with orlistat, the only commonly used anti-obesity drug on the market, we found that they achieved comparable efficacy in rats. The evidence and advantages are more beneficial for CTS and COS to be accepted as anti-obesity supplements or drugs by researchers and the public.

## 4. Materials and Methods

### 4.1. Materials

COST (MW ≤ 1000 Dalton or Da), COSM (MW ≤ 3000 Dalton or Da) (degree of deacetylation, ≥ 90%; lot 160326C and 160408C), and CTS (degree of deacetylation, ≥ 85%; lot 171112A) were obtained from Shangdong AK Biotech Co., Ltd. (Qingdao, Shandong, China). Orlistat was supplied by Zhongshan Wanhan Pharmaceutical Co., Ltd. (Guangzhou, GuangDong, China) (Appendix A). IBMX, dexamethasone, and insulin were purchased from Sigma-Aldrich Trading Co. Ltd. (Shanghai, China).

### 4.2. Animals and Cell Culture

Sprague-Dawley (SD) rats (male, 200 ± 20 g, 8 weeks old) were obtained from the Guangdong Medical Laboratory Animal Center (Guangzhou, China). The animals were fed in a specific pathogen free (SPF) room (22–25 °C, 50–70% relative humidity, 12:12 h light/dark cycle). All animal experimental protocols were approved by the Institutional Animal Care and Use Committee of Guangdong Pharmaceutical University (Guangzhou, China). After one week of adaptive feeding, 10 rats continued to be fed the normal food (NF) diet, and others were fed a high-fat diet (HFD), The composition of HFD was 54% basic feed, 15% lard, 15% sucrose, 4% milk powder, 3% peanut, 5% egg yolk powder, 2% salt, 1% sesame oil, 0.6% CaHPO4, and 0.4% mountain flour (Appendix A). After 8 weeks, the body weight of rats fed a HFD was 20% more than that of rats fed the NF diet, indicating that the obesity model was successfully established.

There were 50 obese rats randomly divided into the following 5 groups (n = 10 per group): The HF group (vehicle control) received a 0.9% NaCl solution via oral gavage; according to a previous study [20,25,48], the orlistat group, COST group, COSM group, and CTS group received orlistat (7 mg/kg·day), COST (600 mg/kg·day), COSM (600 mg/kg·day), CTS (600 mg/kg·day), respectively, for 8 weeks. These rats were continuously fed a HFD. Here, we used orlistat as a positive drug control group.

The 3T3-L1 cell line was purchased from the cell bank of the Chinese Academy of Sciences and stored frozen at Guangdong Pharmaceutical University. The 3T3-L1 cells were rapidly thawed and resuscitated from liquid nitrogen, and cultured in 10% fetal bovine serum (FBS) Dulbecco Modified Eagle Medium (DMEM) complete medium at 37 °C, 5% CO2 until the cell fusion degree reached 80–90%, subculture, or the next experiment. The cells were cultured for 2 days in a medium containing 0.5 mM IBMX, 1 μM dexamethasone, 10 ug/mL insulin, and 0.1 μL rosiglitazone, and then cultured for 12 days using a medium containing 10 μg/mL insulin. The cells of the model control group were cultured for 14 days in a complete medium containing the above inducer. Cells in the low, medium and high doses of COST (COST-L, -M, -H), and low, medium and high doses of COSM (COSM-L, -M, and -H) treatment groups were administered with different concentrations of COST and COSM while inducing agents. (The concentrations of COST, COSM low, medium, and high doses were 0.5mg/mL, 2.5mg/mL, and 5mg/mL.) (Appendix A).

### 4.3. Records of Food Intake, Bodyweight Serum, and Cellular Biochemical Analysis

The daily food intake and weekly weight changes of the rats were recorded. All rats were free to eat and drink ad libitum. After eight weeks of treatment, the rats were fasted for 24 h and then anesthetized, and blood samples were collected from the abdominal aorta. We quickly harvested the liver, WAT, and BAT of rats and weighed them. After quickly freezing in liquid nitrogen, the tissue was stored at −80 °C for further experiments. After cell induction and administration, the cell culture medium was collected for glucose measurement; the remaining cells were washed with phosphate buffer saline (PBS), ultrasonically homogenized, and the homogenate was used to determine the TG content. The levels of TG, TC, LDL-C, HDL-C, FFA, and glucose in serum and cell were determined using commercially available kits purchased from Nanjing Jianchneg Bioengineering Institute, Inc. (Nanjing, China).

### 4.4. Metabolic Measurements

The test was carried out using the metabolic detection system CLAMS (Columbus Instruments’ Comprehensive Lab Animal Monitoring System) and examination tool (CLAX, Columbus Instruments International, Columbus, OH, USA). Three SD rats were randomly selected from each group and placed in a metabolic activity cage, one for each cage. All rats were free to eat and drink ad libitum. We set the instrument program and warmed up the machine for 30 min, and checked the sealing of the metabolic cage. Each rat was acclimated into a system capable of simultaneously monitoring as many as 8 subjects for 24 h and then monitored over 48 consecutive h to measure VO_2_ and energy expenditure. The energy metabolism indicators of rats at standard room temperature were monitored and compared. Energy expenditure and the respiratory exchange ratio (RER = VCO_2_/VO_2_) were calculated from the gas exchange data; the data were smoothed to ± 1 data point.

### 4.5. Digital Gene Expression Tag Profiling

Total RNA was extracted from WAT using a commercial kit, and the concentration and quality were detected. The BGI group (Shenzhen, GuangDong, China) conducted digital gene expression tag profiling. The total RNA was treated by mRNA enrichment or rRNA removal to obtain purified RNA. The obtained RNA was fragmented, and double-stranded DNA was synthesized by reverse transcription. The ends of the synthesized double-stranded DNA were filled in and the 5′ end was phosphorylated, and the 3′ end formed an “A” end, followed by a bubbling linker having a convex “T” at the 3′ end. The ligation product was PCR amplified by specific primers. The PCR product was thermally denatured into a single-strand, and the single-stranded DNA was circularized with a bridge primer to obtain a single-stranded circular DNA library. The prepared library was sequenced, and, subsequently, the differentially expressed genes (DEGs) were identified using the NOISeq method. Then, the DEGs were clustered with gene ontology (GO) analysis (Appendix A).

### 4.6. Quantitative RT-PCR

We weighed about 0.5 g of each group of fat, and total RNA was extracted using a commercial kit (TRIzol, Invitrogen, Inc., Carlsbad, CA, USA). Single-stranded complementary DNA (cDNA) was prepared by reverse transcription using a kit (TaKaRa, Lot D413KA5332, Shiga, Otsu, Japan). The cDNA products were amplified using real-time RT-PCR with the TaKaRa SYBR Premix Ex Taq™ kit, a real-time PCR system, and PikoReal analysis software (Thermo Fisher Scientific Oy, Vantaa, Finland) (PCR amplification procedure: Initial denaturation at 95 °C for 30 s, denaturation at 95 °C for 3 s, annealing at 60 °C for 20 s, and extension at 65 °C for 15 s, 40 cycles.). The gene primers used for PCR were designed and synthesized at Sangon Biotech Co. Ltd. (Shanghai, China). Primer pairs include UCP1, PGC1α, PRDM16, p38-MAPK, TMEM26, ATF2, Slc27a1, β-actin, PKA, Dio2, FTO, PPARγ, and PPARα (Appendix A).

### 4.7. Western Blotting

Total protein was extracted from SD rat adipose tissue using Radio-Immunoprecipitation Assay (RIPA) lysis buffer (Dalian Meilun Biotechnology Co., LTD., Dalian, China) containing protease and phosphatase inhibitor (Beyotime, Shanghai, China). Protein samples were diluted using loading buffer and denatured at 98 °C for 5 min. The protein sample was electrophoresed on the sodium dodecyl sulfate-polyacrylamide gel (SDS-PAGE) at a concentration of 5–15% (*v*/*v*), and the isolated protein was subsequently transferred to a polyvinylidene fluoride (PVDF) membrane (Millipore Corp., Billerica, MA, USA). The PVDF membrane was blocked with 5% skimmed milk powder. After blocking, the PVDF membrane was incubated overnight at 4 °C in diluted primary antibody and then incubated with goat anti-rabbit IgG/horseradish peroxidase (HRP) secondary antibody for 1 h (Biosynthetic Biotechnology Co., Ltd., Beijing, China). Primary antibodies include UCP1, PRDM16, PGC-1α, P38-MAPK, ATF2, and β-actin (Abcam plc. shanghai, China) (Appendix A). Gray analysis of Western blots was performed by Image J 1.48V software (National Institutes of Health, USA).

### 4.8. Statistical Analysis

All data are expressed as the mean ± SD. Statistical analysis was performed by one-way ANOVA using the GraphPad Prism 7.0 software (GraphPad Software, Inc., La Jolla, CA, USA). A P value < 0.05 was considered to represent a significant difference.

## 5. Conclusions

CTS, COST, and COSM can accelerate the browning of WAT and thermogenesis of BAT by up-regulating the gene expression of UCP1, PRDM16, and PGC-1α, thereby accelerating energy metabolism and increasing fat consumption in obese rats. CTS, COS may be potentially valuable natural products for the prevention and treatment of obesity.

## Figures and Tables

**Figure 1 molecules-24-04455-f001:**
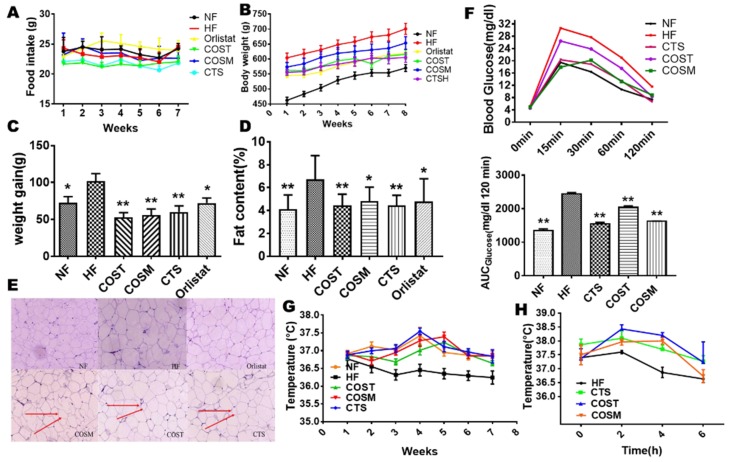
Chitosan (CTS), COST (chitosan oligosaccharide, MW ≤ 1000 Da), and COSM (chitosan oligosaccharide, MW ≤ 3000 Da) reduce weight gain, decrease adipose tissue, and increase thermogenesis. Male Sprague Dawley (SD) rats (8 weeks old) were fed a high-fat diet (HFD) for eight weeks to establish an obesity model. These rats received orlistat (7 mg/kg·day), COST (600 mg/kg·day), COSM (600 mg/kg·day), and CTS (600 mg/kg·day) for 8 weeks. The daily food intake (**A**), weekly body weight (**B**), and body weight gain (**C**) of rats were recorded during the experimental period (n = 10 mice per group). The fat content (**D**) and epididymal fat cells (**E**) in the treatment group were significantly reduced, and their oral glucose tolerance test (OGTT) was significantly improved (**F**). The adjacent bar graphs represent the area under the curve (AUC) of OGTT. Increased rectal temperature in the treated group at normal temperature (22 °C) (**G**) and cold stimulation (4 °C) (**H**) indicated increased thermogenesis. The red arrow indicates the relatively small volume of fat cells. Data are expressed as the mean ± Standard Deviation (SD), * *p* < 0.05 and ** *p* < 0.01 vs. the HF group.

**Figure 2 molecules-24-04455-f002:**
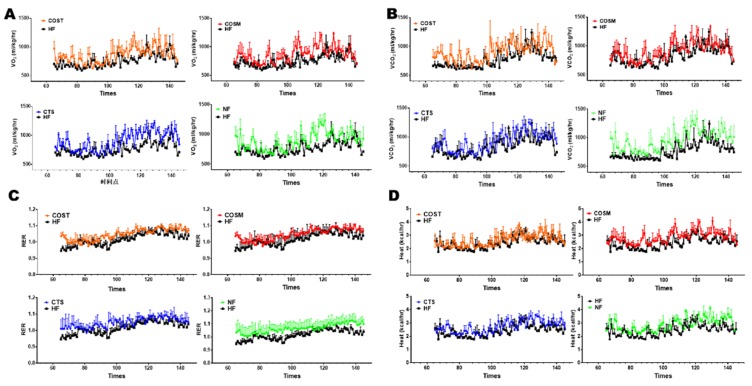
CTS, COST, and COSM increased the energy expenditure of obese rats. Energy expenditure was assessed by measuring the volume of oxygen (VO2) (**A**), volume of carbon dioxide (VCO2) (**B**), respiratory exchange ratio (RER) (**C**), and heat production (**D**) of each group (n = 3) of obese rats at room temperature (22 °C). Data are expressed as the mean ± Standard Deviation (SD), * *p* < 0.05 and ** *p* < 0.01 vs. the HF group.

**Figure 3 molecules-24-04455-f003:**
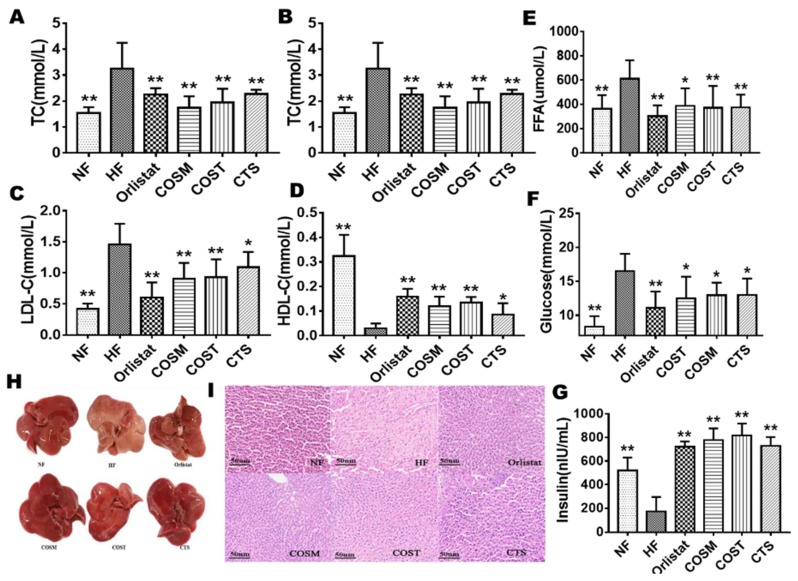
CTS, COST, and COSM improve the blood index and liver fat accumulation of obese rats. Analysis of blood indicators showed that the blood levels of triglyceride (TG) (**A**), total cholesterol (TC) (**B**), LDL-C (**C**), HDL-C (**D**) and FFA (**E**) were significantly improved in the CTS, COST, and COSM treatment groups (n = 10 per group). Serum glucose (**F**) levels were significantly reduced, and serum insulin (**G**) levels were significantly increased in the CTS, COST, and COSM treatment groups (n = 10 per group). (**H**): Appearance of liver tissue in the rats of the NF, HF, orlistat, COSM, COST, and CTS groups. (**I**): HE staining of epididymal WAT in the rats of the NF, HF, orlistat, COSM, COST, and CTS groups. Data are expressed as the mean ± Standard Deviation (SD), * *p* < 0.05 and ** *p* < 0.01 vs. the HF group.

**Figure 4 molecules-24-04455-f004:**
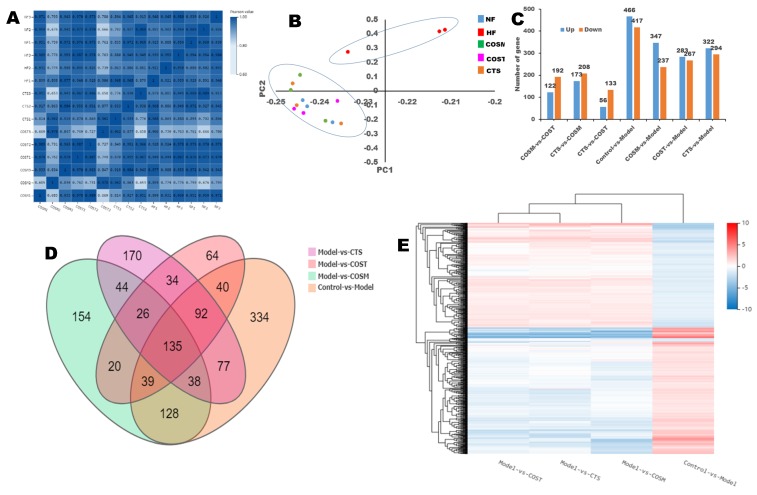
Digital gene expression profiling (DGE) analysis. (**A**) The sample correlation heat map (n = 3 per group), (**B**) principal components analysis (PCA), (**C**) histogram of up- and down-regulated genes, (**D**) Venn diagram of DEGs. Graph of the quantity of DEGs (fold change ≥ 2 and adjusted P value ≥ 0.001), (**E**) clustering heat map of DEGs.

**Figure 5 molecules-24-04455-f005:**
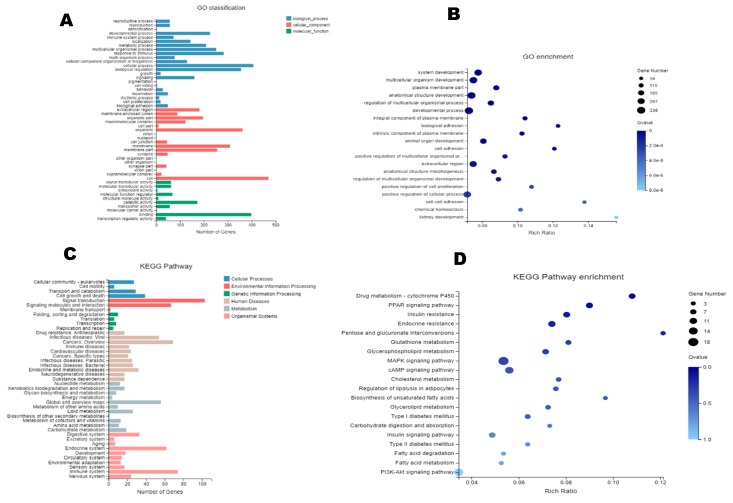
DGE analysis. (**A**): Gene Ontology (GO) annotation classification of DEGs. (**B**): GO enriched bubble map of DEGs (FDR ≤ 0.01). (**C**): KEGG pathway annotation of DEGs. (**D**): Enriched bubble map of the Kyoto Encyclopedia of Genes and Genomes (KEGG) pathway analysis of DEGs (FDR ≤ 0.01).

**Figure 6 molecules-24-04455-f006:**
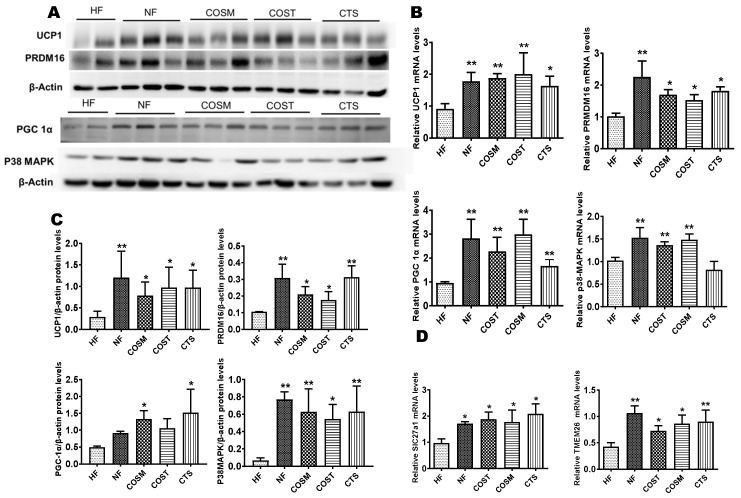
CTS, COST, and COSM increase the expression of browning genes and proteins in epididymal WAT. **A–C**: Expression of brown adipocyte-marker genes and proteins, qRT-PCR (**B**) and Western blotting (**A**) to quantify the relative mRNA and protein expression levels of PRDM16, UCP1, PGC1α, and p38-MAPK. Grayscale analysis (**C**) of Western blots was performed by ImageJ software. Slc27a1 and TMEM26 genes were quantified by qRT-PCR (**D**). Data are expressed as the mean ± Standard Deviation (SD), * *p* < 0.05 and ** *p* < 0.01 vs. the HF group.

**Figure 7 molecules-24-04455-f007:**
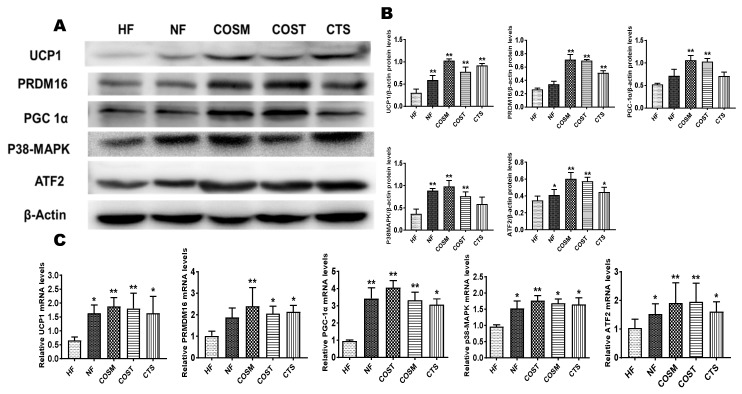
CTS, COST, and COSM increase the expression of browning genes and proteins in brown adipose tissue (BAT). **A–C**: Expression of brown adipocyte-marker genes and proteins, qRT-PCR (**C**) and Western blotting (**A**) to quantify the relative mRNA and protein expression levels of PRDM16, UCP1, PGC1α, and p38-MAPK. Grayscale analysis (**B**) of Western blots was performed by ImageJ software. Data are expressed as the mean ± Standard Deviation (SD), * *p* < 0.05 and** *p* < 0.01 vs. the HF group.

**Figure 8 molecules-24-04455-f008:**
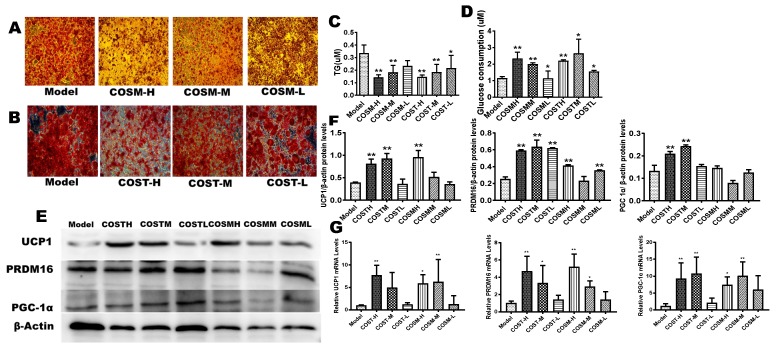
COST and COSM inhibit the differentiation of 3T3-L1 preadipocytes into mature adipocytes and induce the expression of browning genes and thermogenic genes. The logarithmic growth phase of 3T3-L1 preadipocytes was induced by culturing for 2 days in medium containing 0.5 mM IBMX, 1 μM dexamethasone, 10 ug/mL insulin, and 0.1 μL rosiglitazone. The cells were cultured for 12 days using medium containing 10 µg/mL insulin. A, B: Effect of COSM (**A**) and COST (**B**) on the differentiation of 3T3-L1 preadipocytes. (**C**): Levels of TGs in 3T3L1 preadipocytes in each group. (**D**): Levels of glucose in the culture medium of 3T3L1 cells in each group. The qRT-PCR (**G**) and Western blotting (**E**) were used to quantify the relative mRNA and protein expression levels of PRDM16, UCP1, and PGC1α. Grayscale analysis (**F**) of Western blots was performed by ImageJ software. Data are expressed as the mean ± Standard Deviation (SD), * *p* < 0.05 and ** *p* < 0.01 vs. the HF group.

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
