# Peer review of "Beneficial Metabolic Effects of Chitosan and Chitosan Oligosaccharide on Epididymal WAT Browning and Thermogenesis in Obese Rats"

_molecules, 2019, doi:10.3390/molecules24244455_

Round 1
Reviewer 1 Report
The present study about the effects of CTS and COS in obesity is very well designed, very complete and demonstrate an incredible amount of work that extensively describe the mechanism of action of CTS and COS in the context of obesity. Overall, the manuscript seems to me well written and understandable. However a minor English revision would be necessary in all sections.
My concerns about the present paper are related with its novelty and its scientific impact. So please discuss better how what does your study brings new comparing to other studies that are also proposing CTS and COS as anti-obesity supplements? And are the advantages that of this substances enough to compete with the huge amount of other anti-obesity supplements in the market?
I also recommend to do a second revision to the all manuscript to check for spelling error and repeated sentences (as an example, sentence 334 and 339" in summary..." are the same)
The quality of figure 5 as well as some of the western blot images should be improved.
Author Response
Dear Reviewer:
Thanks for your comments and for concerning our manuscript entitled “Beneficial Metabolic Effects of Chitosan and Chitosan Oligosaccharide on Epididymal WAT Browning and Thermogenesis in Obese Rats”. These comments are all valuable and very helpful for revising and improving our manuscript. They also have important guiding significance for our further research. We have studied comments carefully and have made correction which are listed below point by point. And we hope that the revised manuscript could meet with your approval.
We have described the experimental method in more detail. Some tables and pictures of the experimental method are also included in the supplementary data. See section 4 for more information. Modified parts in the text are in red font.
Point 1: The present study about the effects of CTS and COS in obesity is very well designed, very complete and demonstrate an incredible amount of work that extensively describe the mechanism of action of CTS and COS in the context of obesity. Overall, the manuscript seems to me well written and understandable. However a minor English revision would be necessary in all sections.
Response 1: After the manuscript was revised, we carefully modified it.
Point 2: My concerns about the present paper are related with its novelty and its scientific impact. So please discuss better how what does your study brings new comparing to other studies that are also proposing CTS and COS as anti-obesity supplements? And are the advantages that of this substances enough to compete with the huge amount of other anti-obesity supplements in the market?
Response 2: Thank you for your suggestion. Based on reviewers' comments, we discussed that the study brings new compared to other studies that are also proposing CTS and COS as anti-obesity supplements. As following:
“Previous anti-obesity effects on COS and CTS were primarily focused on their effects on leptin resistance and fat formation [1-3]. Recent research reports show that browning of WAT and increased thermogenesis are effective against obesity [4, 5], and there are few studies on COS and CTS in promoting WAT browning and increasing thermogenesis. This study mainly explored the role and specific mechanism of COS and CTS to promote WAT browning and increase thermogenesis. Nowadays, the potential side effects or adverse drug reactions of anti-obesity drugs on the market have gradually become a major public health problem, which is also a major obstacle to the development of new drugs [6]. Therefore, natural products with low toxicity and anti-obesity effects have become popular trends in anti-obesity drugs or supplements. As a safe and non-toxic anti-obesity natural product, CTS and COS have many advantages, such as wide source, high yield, good biocompatibility and economical and practical [7-9]. This study further clarified the role and mechanism of COS and CTS against obesity. By comparing with Orlistat, the only commonly used anti-obesity drug on the market, we found that they achieved comparable efficacy in rats. These evidences and advantages are more beneficial for CTS and COS to be accepted as anti-obesity supplements or drugs by researchers and the public.”
Point 3: I also recommend to do a second revision to the all manuscript to check for spelling error and repeated sentences (as an example, sentence 334 and 339" in summary..." are the same)
Response 3: Thank you and sorry for our negligence. We have commissioned a professional agency for further careful spelling and grammar checking of manuscripts.
Point 4: The quality of figure 5 as well as some of the western blot images should be improved.
Response 4: Thank you and sorry for our negligence. We have updated the images in the manuscript to use high definition images.
References:
1.Pan H, Fu C, Huang L, Jiang Y, Deng X, Guo J, et al. Anti-Obesity Effect of Chitosan Oligosaccharide Capsules (COSCs) in Obese Rats by Ameliorating Leptin Resistance and Adipogenesis. Marine drugs. 2018;16(6):198.
2.S T, B G, Y T, J G, ZQ S. Antiobese effects of capsaicin-chitosan microsphere (CCMS) in obese rats induced by high fat diet. Journal of agricultural and food chemistry. 2014;62(8):1866-74.
3.HL Z, XB Z, Y T, SH W, ZQ S. Effects of chitosan and water-soluble chitosan micro- and nanoparticles in obese rats fed a high-fat diet. International journal of nanomedicine. 2012;7(undefined):4069-76.
4.Scheele C, Wolfrum C. Brown adipose cross talk in tissue plasticity and human metabolism. Endocrine reviews. 2019.
5.Villarroya F, Cereijo R, Villarroya J, Giralt M. Brown adipose tissue as a secretory organ. Nat Rev Endocrinol. 2017;13(1):26-35.
6.Sun N-N, Wu T-Y, Chau C-F. Natural Dietary and Herbal Products in Anti-Obesity Treatment. Molecules (Basel, Switzerland). 2016;21(10):1351.
7.Naveed M, Phil L, Sohail M, Hasnat M, Baig M, Ihsan AU, et al. Chitosan oligosaccharide (COS): An overview. Int J Biol Macromol. 2019;129:827-43.
8.Younes I, Rinaudo M. Chitin and chitosan preparation from marine sources. Structure, properties and applications. Marine drugs. 2015;13(3):1133-74.
9.Sharif R, Mujtaba M, Ur Rahman M, Shalmani A, Ahmad H, Anwar T, et al. The Multifunctional Role of Chitosan in Horticultural Crops; A Review. Molecules (Basel, Switzerland). 2018;23(4):872.
Reviewer 2 Report
The present manuscript is focused on the study of the efficacy of chitosan and of a metabolite subrogated (Chitosan oligosaccharide) as an antiobesity agent. The study is well designed, and the methodology applied is adequate. Likewise, most of the results are well detailed and support the conclusion obtained. However, there are several important parts of the results should be more detailed. Also, there are several points that authors should be revised previously to be accept the manuscript:
Minor comment
Abstact. The first sentence of the abstract could be improved and change the order starting with the importance of investigating anti-obesity drugs instead of “This present experiment...”
The quality of figures 4 and 5 will be revised. Figures are too small.
Supplementary data is not available?? Histopathology is not described in main text and I haven´t access to suppl data to check it.
A briefly description of bioinformatic analysis carried out will be include in the main text and/or implemented the statistical analysis section.
Mayor comments
It would be recommended to explain in more detail, if it known, the importance of the different derivatives of COS supplements (COST/COSM and CTS) that are going to be used in the manuscript, beyond the differences in molecular weight; and that, evidence the importance of studying them all. In fact, due to no significantly changes were detected between treatments, it could be interesting evidences the mayor or minor presence (or abundance) of these derivates. Both products (COST and COSM) are obtained in equal proportion? Do all the components used have the same benefits?
Why authors use orlistat in the experimental design?? A shortly description of orlistat will be necessary to understand why authors include the study of this molecule together COS supplements. Since the authors include the use of orlistat in the comparative it would be interesting to show whether these natural products have a greater effect than this reference molecule to treat obesity.
Differential Gene Expression between treatments will be more described. The figure 4 is poorly described. Specific differences in the DEGs among the groups (COST, COSM, CTS and HF groups) will be more describe.
Authors have enough data to deepen in the mechanism involved in the effect of chitosan and chitosan oligosaccharide on weight loss i.e. by expanding the studies including in a second set of analysis, if there is any difference between CTS and COS in order to clarify the use of the component that has the greatest benefit for the treatment of obesity or, using correlation studies with plasma parameters obtained. Moreover, a comparative with orlistat will be useful to promote (or not) the role of these molecules in obesity. Thus, the main objective of this review would be to determine whether there would be any component of those studied that had a greater impact on the treatment of obesity.
Author Response
Dear Reviewer:
Thanks for your comments and for concerning our manuscript entitled “Beneficial Metabolic Effects of Chitosan and Chitosan Oligosaccharide on Epididymal WAT Browning and Thermogenesis in Obese Rats”. These comments are all valuable and very helpful for revising and improving our manuscript. They also have important guiding significance for our further research. We have studied comments carefully and have made correction which are listed below point by point. And we hope that the revised manuscript could meet with your approval.
We have described the experimental method in detail. Some tables and pictures of the experimental method are also included in the supplementary data. For related information, please see Section 4. The description of the current status of anti-obesity drugs has also been added in the "Introduction" section. Modifications in the text are in red font.
Minor comment
Point 1: Abstract. The first sentence of the abstract could be improved and change the order starting with the importance of investigating anti-obesity drugs instead of “This present experiment...”
Response 1: Reviewer gave a great suggestion, so we modified this section and provided the importance of investigating anti-obesity drugs. As following:
“Overweight and obesity have become a worldwide health problem and the prevalence is rising. In the last few decades, many anti-obesity chemicals have withdrawn from the market due to serious adverse reactions, and many researchers have turned their attention to low-toxic natural products.”
Point 2: The quality of figures 4 and 5 will be revised. Figures are too small.
Response 2: Thank you and sorry for our negligence. We have updated the images in the manuscript to use high definition images.
Point 3: Supplementary data is not available?? Histopathology is not described in main text and I haven´t access to suppl data to check it.
Response 3: We are sorry that the reviewers have not received the Supplementary data, but we uploaded the relevant Supplementary data (PDF).
We are grateful for the reviewers' suggestions for the insufficient description of Histopathology. And we added a description of Histopathology as following:
“Compared with the NF group, the liver cells of the HF group were deformed, the size was not uniform, and there were more fat vacuoles; the fat vacuoles in the liver of the Orlistat, COST, COSM and CTS treatment groups were significantly reduced compared to the HF group (Figure 3I).”
Point 4: A brief description of bioinformatic analysis carried out will be included in the main text and/or implemented the statistical analysis section.
Response 4: At the request of the reviewers, we described and analyzed the bioinformatics analysis to support our conclusions. Please see section 2.3 for details.
Mayor comments
Point 1: It would be recommended to explain in more detail, if it known, the importance of the different derivatives of COS supplements (COST/COSM and CTS) that are going to be used in the manuscript, beyond the differences in molecular weight; and that, evidence the importance of studying them all. In fact, due to no significantly changes were detected between treatments, it could be interesting evidences the mayor or minor presence (or abundance) of these derivates. Both products (COST and COSM) are obtained in equal proportion? Do all the components used have the same benefits?
Response 1: Thanks for your suggestion. Our answer is as following:
“It is generally believed that the degree of polymerization (DP), the degree of deacetylation (DD) and the molecular weight (MW) are the principal characteristics and key factors of COS [1-3]. COS with a higher DP (≥6), a lower DD and a lower MW has a stronger biological activity than COS with a lower DP, a higher DD and a higher MW [2, 3]. The DP of COS with glucosamine as the monomer is 2-20, and the maximum MW of COS is less than 4000 [3]. The MW of CTS is generally higher than 4000. Theoretically, the MW of COS with the strongest biological activity is 1000-3000. Therefore, this study selected COST (MW≤1000), COSM (MW≤3000) and CTS (MW>4000) as the research objects and their DDs are higher than 90%. COST, COSM, and CTS represent three molecular weight COS derivatives, which researchers focus more on.”
“However, the anti-obesity effects of COST, COSM and CTS in this study did not show significant differences. We conclude that the reasons for this result are mainly: 1) the MW ranges of the two COSs and CTS in this experiment had a region of common overlap; 2) the DP of COSs and CTS used in this study is not and cannot be controlled; 3) the active substances or groups that are effective in COST, COSM and CTS may be the same or similar. The complex structure of COS determines the difficulty of its biological evaluation, and it is one-sided and difficult to evaluate its anti-obesity activity based only on the molecular weight of COS. Therefore, the next step may be to study the specific anti-obesity effect of COSs with specific DP.”
“COST (MW≤1,000 Da), COSM (MW≤3,000 Da) (degree of deacetylation, ≥90%; lot: 160326C and 160408C) and CTS (degree of deacetylation, ≥85%; lot: 171112A) were obtained from Shangdong AK Biotech Co., Ltd. (Qingdao, Shandong, China). The food-grade COST and COSM that we used were obtained through the same and strict biotechnology.”
Point 2: Why authors use orlistat in the experimental design?? A shortly description of orlistat will be necessary to understand why authors include the study of this molecule together COS supplements. Since the authors include the use of orlistat in the comparative it would be interesting to show whether these natural products have a greater effect than this reference molecule to treat obesity.
Response 2: This is a great suggestion. We choose orlistat mainly for comparison with the effects of COS and CTS. Orlistat is one of the best anti-obesity drugs on the market today. The comparison with orlistat facilitates our preliminary assessment of the activity and market value of COS and CTS. Therefore, we have described orlistat according to the opinions of reviewers:
1)In "Introduction" we describe the current use of anti-obesity drugs and the status of orlistat in these anti-obesity drugs (section 1);
2)In the "results" section involving orlistat, we made a descriptive comparison (section 2.1 and 2.2);
3)In "Discussion", we further analyzed that to study the anti-obesity efficacy of COS and CTS, it is meaningful to compare with orlistat (section 3).
Point 3: Differential Gene Expression between treatments will be more described. The figure 4 is poorly described. Specific differences in the DEGs among the groups (COST, COSM, CTS and HF groups) will be more describe.
Response 3: Thanks for the reviewer’s suggestion. We specifically describe Figure 4 to discuss the HFD-induced genetic differences in rats, and the results show that COST, COSM, and CTS can suppress this genetic difference (Section 2.3).
References:
1.EJ C, MA R, SW K, YM B, HJ H, JY O, et al. Chitosan oligosaccharides inhibit adipogenesis in 3T3-L1 adipocytes. Journal of microbiology and biotechnology. 2008;18(1):80-7.
2.Liaqat F, Eltem R. Chitooligosaccharides and their biological activities: A comprehensive review. Carbohydrate Polymers. 2018;184:243-59.
3.Naveed M, Phil L, Sohail M, Hasnat M, Baig M, Ihsan AU, et al. Chitosan oligosaccharide (COS): An overview. Int J Biol Macromol. 2019;129:827-43.
Round 2
Reviewer 2 Report
I have received an appropriate response to all the comments proposed. Congratulations.